# The Pivotal Role of Galectin-3 in Viral Infection: A Multifaceted Player in Host–Pathogen Interactions

**DOI:** 10.3390/ijms24119617

**Published:** 2023-06-01

**Authors:** Bojana S. Stojanovic, Bojan Stojanovic, Jelena Milovanovic, Aleksandar Arsenijević, Milica Dimitrijevic Stojanovic, Nebojsa Arsenijevic, Marija Milovanovic

**Affiliations:** 1Center for Molecular Medicine and Stem Cell Research, Faculty of Medical Sciences, University of Kragujevac, 34000 Kragujevac, Serbia; bojana.stojanovic04@gmail.com (B.S.S.); jelenamilovanovic205@gmail.com (J.M.); aleksandar@medf.kg.ac.rs (A.A.); milicadimitrijevic@yahoo.com (M.D.S.); arne@medf.kg.ac.rs (N.A.); marijaposta@gmail.com (M.M.); 2Department of Pathophysiology, Faculty of Medical Sciences, University of Kragujevac, 34000 Kragujevac, Serbia; 3Department of Surgery, Faculty of Medical Sciences, University of Kragujevac, 34000 Kragujevac, Serbia; 4Department of Histology, Faculty of Medical Sciences, University of Kragujevac, 34000 Kragujevac, Serbia; 5Department of Pathology, Faculty of Medical Sciences, University of Kragujevac, 34000 Kragujevac, Serbia

**Keywords:** galectin-3, viral infection, host–pathogen interactions, inflammation, therapeutic target

## Abstract

Galectin-3 (Gal-3), a beta-galactoside-binding lectin, plays a pivotal role in various cellular processes, including immune responses, inflammation, and cancer progression. This comprehensive review aims to elucidate the multifaceted functions of Gal-3, starting with its crucial involvement in viral entry through facilitating viral attachment and catalyzing internalization. Furthermore, Gal-3 assumes significant roles in modulating immune responses, encompassing the activation and recruitment of immune cells, regulation of immune signaling pathways, and orchestration of cellular processes such as apoptosis and autophagy. The impact of Gal-3 extends to the viral life cycle, encompassing critical phases such as replication, assembly, and release. Notably, Gal-3 also contributes to viral pathogenesis, demonstrating involvement in tissue damage, inflammation, and viral persistence and latency elements. A detailed examination of specific viral diseases, including *SARS-CoV-2*, *HIV*, and *influenza A*, underscores the intricate role of Gal-3 in modulating immune responses and facilitating viral adherence and entry. Moreover, the potential of Gal-3 as a biomarker for disease severity, particularly in COVID-19, is considered. Gaining further insight into the mechanisms and roles of Gal-3 in these infections could pave the way for the development of innovative treatment and prevention options for a wide range of viral diseases.

## 1. Introduction

Viruses are diverse and intricate pathogens that have evolved to infect a wide range of hosts, including humans [1]. They are classified based on their genomic properties, dividing them into DNA and RNA viruses [2]. To date, 219 virus species have been identified as capable of infecting humans [3]. The yellow fever virus, an RNA virus, was the first human virus to be described in 1901 by Reed and Carroll [3]. Interestingly, more than two-thirds of viruses that infect humans can also infect other animals, and many emerging viruses that infect humans have been found to originate from mammals or birds [4]. In the complex interplay between viruses and their hosts, Gal-3 has emerged as a multifaceted player in host–pathogen interactions, playing a pivotal role in viral infections [5]. This review will explore the various roles of Gal-3 in viral infections and its potential implications for host immunity and antiviral strategies.

Lectins are proteins that bind to specific sugar moieties or structures and play essential roles in various cellular and physiological regulations [6]. Galectins, a type of S-type lectin, bind to β-galactosides, such as disaccharide N-acetyllactosamine, found in N-linked and O-linked glycoproteins [7]. Galectins are involved in regulating mRNA splicing, apoptosis, cell cycle, cancer formation, metastasis, immune responses, and more [8]. They are considered pattern recognition receptors and are found in various immune cells and tissues, where they play potential and significant roles in human infections caused by different types of pathogens [7,9].

Galectin-3 is a unique chimera-type galectin characterized by a C-terminal carbohydrate recognition domain (CRD) and a large N-terminal (NT) protein-binding domain (Figure 1) [10]. Galectin-3 is encoded by the *LGALS3* gene and is primarily expressed in the cell cytoplasm, but it can also move into the nucleus and be secreted onto the cell surface [10,11]. It is involved in various biological activities, including cell growth, pre-mRNA splicing, differentiation, angiogenesis, inflammation, fibrosis, apoptosis, and host defense [12]. Galectin-3 is expressed abundantly during viral infections in various immune cells, fibroblasts, and epithelial and endothelial cells [12]. It not only regulates viral entry and attachment but also mediates inflammatory responses, potentially causing an imbalanced production of pro-inflammatory cytokines [13].

## 2. The Comprehensive Impact of Galectin-3 on Viral Infections: From Entry to Immunity, Life Cycle, and Disease Progression

Galectin-3 is a multifaceted protein that plays a crucial role in various aspects of viral infections, with diverse and sometimes contrasting effects depending on the virus and the host’s immune response [14]. This section aims to provide a comprehensive understanding of the impact of Gal-3 on viral infections, encompassing its involvement in viral entry, modulation of immune responses, regulation of the viral life cycle, and its contribution to disease progression (Figure 2). In Table 1, we present an overview of the distinct roles of Gal-3 in different viral infections. Through examining the complex interplay between Gal-3 and different viral pathogens, we seek to elucidate the mechanisms through which Gal-3 can either promote or inhibit infection, as well as its potential as a therapeutic target for the treatment of various viral diseases.

### 2.1. Galectin-3 and Viral Entry

Viral entry is a crucial step in the infection process, where the virus gains entry into the host cell to replicate and further spread the infection. This process usually involves the interaction between viral proteins and host cell receptors [25]. As a carbohydrate-binding protein, Gal-3 can interact with glycans present on the surface of both host cells and viral particles, which can modulate the viral entry process [26].

#### 2.1.1. Galectin-3 as a Facilitator of Viral Attachment

The initial step in the process of viral infection is viral attachment, as it enables the virus to bind to specific receptors on the surface of host cells and initiate the process of viral replication [25]. Galectin-3, a member of the galectin family of proteins, has a carbohydrate recognition domain (CRD) that binds to β-galactosides found on host cells and viral glycoproteins [26]. In the case of *human immunodeficiency virus* (*HIV*), Gal-3 interacts with the viral envelope glycoprotein gp120, enhancing its binding to host cell surface receptors, such as CD4, and subsequently enhancing infection [15]. Additionally, Gal-3 plays a vital role in the entry of *severe acute respiratory syndrome coronavirus 2* (*SARS-CoV-2*) into host cells, and blocking it may prevent disease progression. Human Gal-3 has significant structural and sequence similarity to the N-terminal domain (NTD) of *SARS-CoV-2*, important for viral entry, suggesting that Gal-3 inhibitors targeting regions with structural overlap to the NTD may have double binding capabilities, providing a potential strategy to inhibit viral entry [16].

Galectin-3 has been identified as a crucial mediator in the entry and attachment of *herpes simplex virus* (*HSV*) during ocular infections and facilitating influenza binding to airway epithelial cells [17,18]. Specifically, *HSV-1* directly binds human Gal-3, and targeted disruption of Gal-3 impairs *HSV-1* infectivity in human corneal keratinocytes [17]. In the case of influenza, Gal-3 binds to the hemagglutinin (HA) protein of *Influenza A viruses (IAVs)* and desialylated airway epithelial cells, increasing *Streptococcus pneumoniae* adhesion [18]. These findings suggest that Gal-3 plays a critical role in viral attachment and entry for *HSV-1* and influenza, making it a potential target for therapeutic intervention.

#### 2.1.2. Galectin-3 and Viral Internalization

Following attachment, Gal-3 may also be involved in the internalization of viruses into host cells. Evidence suggests that Gal-3 can facilitate endocytosis and macropinocytosis, two common routes for viral entry [27]. For example, Gal-3 has been shown to promote *HIV-1* internalization through interacting with the host CD4 receptor and the viral gp120 glycoprotein [15].

Overall, Galectin-3’s ability to interact with viral glycoproteins and host cell surface receptors highlights its potential as a target for antiviral therapies. Through inhibiting Gal-3 function or blocking its interaction with viral glycoproteins, it may be possible to disrupt viral attachment and entry, thus preventing infection and disease progression. Further research is needed to better understand the specific molecular mechanisms underlying Gal-3-mediated viral attachment and to develop targeted therapies to combat various viral infections.

### 2.2. Galectin-3 and Immune Responses

Galectin-3 modulates the immune response during viral infections. It can activate or inhibit immune signaling pathways, depending on the virus and the cellular context [28]. In some cases, Gal-3 promotes the production of pro-inflammatory cytokines and chemokines, such as tumor necrosis factor-alpha (TNF-α) and interleukin-6 (IL-6), enhancing the antiviral response [21].

#### 2.2.1. Galectin-3 in Immune Cell Activation and Recruitment

Galectin-3 influences immune cell activation and recruitment during viral infection, modulating the activation of T cells, dendritic cells, macrophages, and neutrophils and affecting their cytokine production and chemotactic responses [10]. Galectin-3 promotes neutrophil migration towards infection sites via binding to specific glycan structures on their surface and recruiting neutrophils to the lungs during influenza and *Streptococcus pneumoniae* co-infection [7,29]. In COVID-19, there was a strong positive correlation between Gal-3 and interleukin-1 beta (IL-1β), as well as a moderate positive correlation between Gal-3, TNF-α, and IL-12, indicating a T-helper 1 (Th1) immune response [22]. Patients with severe COVID-19 had significantly higher percentages of Gal-3^+^ T cells, which may contribute to elevated systemic Gal-3 levels and promote inflammation [22].

In the context of *Hepatitis B virus* (*HBV*) infection, Gal-3 has been suggested to inhibit IL-10, possibly sustaining *HBV* replication and initiating chronic *HBV* infection [23]. Galectin-3 has also been linked to macrophage polarization during *dengue virus* infection, which is associated with infection severity [30]. The modulation of macrophage polarization by Gal-3 can impact the balance between pro-inflammatory and anti-inflammatory responses, influencing the disease outcome [30]. The complex and context-dependent roles of Gal-3 in viral infections highlight its potential as a therapeutic target but also underscore the need for a deeper understanding of its functions across various infectious diseases.

#### 2.2.2. Galectin-3 in the Regulation of Immune Signaling Pathways

Galectin-3 plays a pivotal role in modulating immune signaling pathways during viral infections, either activating or inhibiting these pathways depending on the cellular context, which influences the immune response [28]. One key pathway influenced by Gal-3 is the Nuclear Factor-kappa B (NF-κB) signaling pathway, a critical transcription factor that regulates genes involved in immune and inflammatory responses [31]. Galectin-3 activates NF-κB, leading to the production of pro-inflammatory cytokines and enhancing the antiviral response, which contributes to the cytokine storm observed in severe COVID-19 cases [32,33].

In addition to NF-κB, Gal-3 modulates other signaling pathways such as Janus kinase/signal transducers and activators of transcription (JAK/STAT), extracellular signal-regulated kinase (ERK), and protein kinase B (AKT), which are involved in various cellular processes [34]. During viral infections, Gal-3 can dysregulate these pathways, leading to aberrant immune responses and contributing to disease pathogenesis [21]. Galectin-3 is also an agonist of Toll-like receptor 4 (TLR4) and can regulate the NOD-like receptor family pyrin domain containing 3 (NLRP3) inflammasome, promoting inflammasome assembly and activation, which contributes to enhanced inflammation and tissue damage in viral infections [31,35,36].

Galectin-3 has been implicated in the regulation of suppressor of cytokine signaling 1 (SOCS1) and retinoic acid-inducible gene I (RIG-I) expression during influenza and *Streptococcus pneumoniae* co-infection [21]. Galectin-3 modulates SOCS1 expression, which may contribute to the overall inflammatory response, and can influence RIG-I expression during co-infection, leading to dysregulated expression and release of pro-inflammatory cytokines [21]. Overall, Galectin-3’s complex and context-dependent role in immune signaling pathways highlights the need for understanding its precise functions in various infections to identify its potential as a therapeutic target or biomarker for disease severity.

#### 2.2.3. Galectin-3 in Modulating Apoptosis and Autophagy

Galectin-3 can also influence host cell survival and death mechanisms, such as apoptosis and autophagy [28,37]. Although the anti-apoptotic function of Gal-3 has been widely reported in various pathological processes, its role in virus-infected cells and its potential contribution to viral replication and persistence remain unclear [38]. Understanding the mechanisms through which Gal-3 impacts viral replication and persistence could lead to the development of novel antiviral therapies targeting Gal-3, providing new options for the treatment of viral diseases. Therefore, further research into the involvement of Gal-3 in virus–host interactions is crucial for the development of effective antiviral treatments.

On the other hand, Gal-3 has been implicated in the regulation of autophagy, a cellular process that can promote viral clearance through degrading viral components [37]. Notably, in *adenovirus* infections, Gal-3 has been demonstrated to suppress autophagic activation [39]. These findings highlight the multifaceted and intricate role of Gal-3 in host–virus interactions, particularly in the context of *adenovirus* infections.

### 2.3. Galectin-3 and Viral Life Cycle

Galectin-3, a multifunctional protein with roles in various biological processes, has emerged as an important player in the context of viral infections [10]. In the following sections, we will explore the involvement of Gal-3 in the viral life cycle and its impact on viral replication, assembly, and release. Understanding the complex interactions between Gal-3 and viral pathogens can provide valuable insights into host–pathogen dynamics and potentially inform the development of novel therapeutic strategies.

#### 2.3.1. Galectin-3 in Viral Replication

Galectin-3 is known to interact with viral proteins and host factors to promote viral genome replication. For example, Gal-3 was shown to facilitate *HIV-1* replication via promoting cell fusion through fibronectin and Gal-3-mediated mechanisms [40]. *HIV-1* Tat, an important regulatory protein for viral replication, can also induce the expression of Gal-3, which contributes to the virus’s pathogenesis through promoting replication and modulating immune responses [41]. *Influenza A virus* replication can be inhibited through upregulating Gal-3 expression, which enhances the expression of antiviral genes that inhibit virus replication [19]. Galectin-3 has also been shown to impact the replication of *Enterovirus 71* (EV71) [20]. Galectin-3 may play a role in sustaining *HBV* replication through stimulating the production of cytokines and chemokines via Cluster of Differentiation 98 (CD98) interactions with macrophages [23]. In the context of *SARS-CoV-2*, Gal-3 inhibition could potentially influence viral RNA synthesis [24].

#### 2.3.2. Galectin-3 in Viral Assembly and Release

Galectin-3 may also play a role in the assembly of viral particles and their release from infected host cells [28]. It has been suggested that Gal-3 can promote the budding of viral particles through interacting with viral structural proteins and host cell membrane components [42]. Galectin-3 has been shown to interact with the *HIV-1* Gag protein and promote viral assembly and release. It functions via interacting intracellularly with Gag and the cellular ALG-2-interacting protein X (Alix), which are essential for viral budding and replication when new infectious virions are generated [42]. Additionally, Gal-3 plays a role in regulating virological synapse formation and facilitating intercellular *HIV-1* transfer among CD4^+^ T cells, providing an alternative pathway for *HIV-1* infection regulation [43]. Exosomes derived from *HIV-1*-infected dendritic cells with high Gal-3 expression have been found to facilitate *HIV-1* infection and dissemination through fibronectin and Gal-3-mediated cell fusion [40]. These findings suggest that Gal-3 plays a crucial role in regulating viral assembly and release, virological synapse formation, and exosome-mediated cell-to-cell communication in *HIV-1* infection. Targeting Gal-3 could represent a promising therapeutic strategy against *HIV-1* infection.

### 2.4. Galectin-3 and Viral Pathogenesis

The involvement of Gal-3 in viral pathogenesis is complex and multifaceted [10]. While it can contribute to tissue damage and inflammation in some viral infections, it can also have protective effects in others [7]. For example, Gal-3 knockout mice have been shown to have reduced lung injury and inflammation during *influenza A virus* infection, suggesting that Gal-3 may exacerbate the disease [44].

#### 2.4.1. Galectin-3 in Tissue Damage and Inflammation

Galectin-3 plays a critical role in mediating tissue damage and inflammation during viral infections through its involvement in immune cell activation, cytokine production, and immune signaling pathways [28]. It contributes to the development of inflammation through promoting the migration and infiltration of immune cells to the site of infection, leading to the release of pro-inflammatory cytokines, chemokines, and other mediators that exacerbate local inflammation and tissue injury [7].

For instance, during *influenza A virus* (IAV) infection, Gal-3 interacts with the NLRP3 inflammasome, promoting its assembly and activation, which leads to the production of pro-inflammatory cytokine IL-1β, resulting in enhanced inflammation and lung tissue damage [44]. In severe COVID-19 cases, excessive inflammation or “cytokine storm” contributes to widespread tissue damage and organ failure, with Gal-3 implicated in this process due to its elevated levels correlating with increased pro-inflammatory cytokines [33]. Galectin-3 also plays a key role in pulmonary-associated lung fibrosis in COVID-19, with positive correlations found between Gal-3 and various markers of inflammation, endothelial injury, and tissue injury [33,45].

Moreover, Gal-3 has been suggested to contribute to fibrogenesis in *HBV* infection through inhibiting IFN-γ secretion and sustaining *HBV* replication, ultimately leading to liver tissue damage and the development of chronic *HBV* infection [14,23]. Galectin-3’s role in mediating tissue damage and inflammation highlights its importance in the complex interplay between the host immune system and viral pathogens during various infections.

#### 2.4.2. Galectin-3 in Viral Persistence and Latency

Galectin-3 has been implicated in the establishment and maintenance of viral persistence and latency [46]. Thus, Gal-3 can modulate host immune responses to promote viral latency, as observed in *HIV* infection [47]. In latently infected cells, the expression of Gal-3 was found to be closely correlated with that of Tat, a crucial protein for *HIV-1* transcriptional activation [47]. The colocalization and possible interaction between Gal-3 and Tat suggest that Gal-3 may play an essential role in Tat’s function within the cells, offering a potential new target for therapeutic strategies to disrupt *HIV-1* latency or prevent viral reactivation in infected individuals.

#### 2.4.3. Galectin-3 and Host Protection

Galectin-3 can sometimes have a protective role during viral infections, promoting antiviral immunity and limiting viral spread, which contributes to host protection and recovery [34]. For instance, in the context of *flavivirus* infection, Gal-3 has been found to have a hepatoprotective function in patients with *Hepatitis C Virus* (*HCV*) and End Stage Renal Disease (ESRD). Higher levels of Gal-3 and IL-6 may protect against ongoing proinflammation and reduce chronic inflammation in these patients [48].

Moreover, Gal-3 has been shown to play a regulatory role in co-infections, such as *influenza A* viruses and *Streptococcus pneumoniae*, through dysregulating the expression of pro-inflammatory cytokines and downregulating SOCS1 expression, thus modulating the immune response and protecting the host from excessive inflammation and tissue damage [21]. In a study examining murine *cytomegalovirus* (*MCMV*)-induced hepatitis, Gal-3 was found to play a protective role, with Gal-3-deficient mice showing higher liver damage, increased serum levels of ALT, and higher virus titers, while treatment with exogenous Gal-3 alleviated *MCMV*-induced liver damage [49].

### 2.5. Galectin-3 Interactions with Sulfated Glycosaminoglycans and Implications for Viral Pathogenesis

Glycosaminoglycan-binding proteins (GAGBPs) and lectins, two primary categories of glycan-binding proteins, interact with both free and proteoglycan-bound glycosaminoglycans (GAGs) [50]. GAGBPs, including growth factors, cytokines, morphogens, and extracellular matrix proteins, are critical regulatory factors in numerous cellular and extracellular processes such as cell growth, metastasis, morphogenesis, neural development, and inflammation [51]. In contrast to lectins, which primarily bind to terminal sugar residues of N- and O-glycans and glycolipids, GAGBPs differentiate themselves through interacting with internal residues of sulfated GAG chains [52].

Among these, Gal-3, a β-galactoside-binding lectin, has demonstrated affinity for N-acetyllactosamine residues on glycoconjugates [52]. Interestingly, it shares features with GAGBPs, including its interaction with unmodified heparin, chondroitin sulfate-A (CSA), -B (CSB), and -C (CSC), as well as chondroitin sulfate proteoglycans (CSPGs) [52]. Furthermore, Gal-3 exhibits significant affinity for desulfated GAGs containing N-acetyl-D-galactosamine (GalNAc), namely chondroitin and dermatan, while no such affinity is observed for hyaluronan and N-acetylheparosan [53]. The ability of Gal-3 to bind GAGs and CSPGs positions this lectin as a potential collaborator or competitor of other GAGBPs, including growth factors, cytokines, morphogens, and extracellular matrix proteins [52].

The dynamic changes in Gal-3 binding to chondroitin 4-sulfate, brought about through a decrease in N-acetylgalactosamine 4-sulfatase activity, have been examined in recent research [54]. This decrease in enzyme activity affects the functionality of Arylsulfatase B (ARSB), a sulfohydrolase responsible for the removal of 4-sulfate groups from N-acetylgalactosamine 4-sulfate residues [54]. ARSB’s action is essential for degrading GAG chains, specifically chondroitin 4-sulfate (C4S) and dermatan sulfate (DS). Through this process, ARSB alters the release and binding of critical molecules, such as Gal-3. Additionally, ARSB’s role in influencing chondroitin 4-sulfate’s sulfation pattern sheds light on new dimensions of intracellular signaling and cell-extracellular matrix interactions [54]. Increasing chondroitin 4-sulfation induces cellular processes, such as transcriptional events, mainly through reducing Gal-3 binding and increasing Gal-3’s availability to interact with transcription factors and other mediators [54].

Understanding the mechanistic approaches and intricate interactions of Gal-3 with sulfated GAGs and their associated proteoglycans is essential. This is particularly critical in evaluating the nature of interactions between viruses and Gal-3, considering Gal-3’s multifaceted role in viral pathogenesis. A detailed focus on this realm not only deepens our knowledge of cellular processes but also opens up new avenues for developing antiviral interventions and therapeutic strategies, an area of research that warrants extensive exploration. Therefore, incorporating findings from in-depth studies that explore the impact of changes in GAG sulfation on galectin binding will significantly enhance our understanding of Gal-3’s interactions and its overall role in health and disease.

### 2.6. Signaling Pathways Triggered by Galectin-3 in Viral Infections

Galectin-3, a well-established participant in an array of cellular activities, has lately been identified as a significant orchestrator of various transcriptional pathways during viral infections. The role of Gal-3 in influencing key signaling pathways integral to the host’s defense mechanisms against viral pathogens is becoming increasingly clear, thus spotlighting its potential in therapeutic interventions.

NF-κB Pathway: One major pathway activated by Gal-3 during viral infections is the NF-κB pathway. Galectin-3 has been found to stimulate the activation of the NF-κB signaling pathway, a key regulator of immune and inflammatory responses [31]. This pathway is critical for the production of pro-inflammatory cytokines and chemokines, aiding in the recruitment and activation of immune cells to the site of infection [55].

TLR Pathway: In addition, Gal-3 has been demonstrated to interact with Toll-like receptors (TLRs), a class of proteins that play a crucial role in the innate immune system [10]. Galectin-3 can bind to TLRs, particularly TLR4, triggering downstream signaling that culminates in the activation of NF-κB and the production of pro-inflammatory cytokines [31]. This function suggests a role for Gal-3 in enhancing innate immune responses to viral infections.

JAK/STAT Pathway: Finally, evidence points to Gal-3 involvement in the Janus kinase/signal transducer and activator of transcription (JAK/STAT) pathway [56]. This pathway is crucial in transducing signals for various cytokines and growth factors, and its activation can lead to an antiviral immune response [57].

These pathways indicate the critical role of Gal-3 in modulating the cellular response to viral infections. However, our understanding of the precise mechanisms of action is still growing. Future research is necessary to better elucidate how Gal-3 regulates these pathways and contributes to the host immune response during viral infections.

### 2.7. Galectin-3: Receptor Interactions and Their Implications for Viral Infections

Galectin-3, possessing a unique carbohydrate recognition domain (CRD), exhibits extensive interaction with a variety of receptors and glycoconjugates [58]. These interactions participate in diverse physiological and pathological processes, reflecting the multifunctional role of Gal-3. Here, we highlight some notable interactions of Gal-3:Insulin Receptor: Gal-3’s involvement in insulin receptor signaling might influence conditions such as type 2 diabetes [59,60].EGFR: Interaction with Epidermal Growth Factor Receptor (EGFR) could implicate Gal-3 in the regulation of cell growth and survival, thus influencing oncogenesis [61].Integrins: Through binding to various integrins, Gal-3 participates in modulating cell adhesion and migration [62].CD45: Interactions with CD45 could allow Gal-3 to adjust immune responses [63].CD71: Through interaction with CD71, Gal-3 plays a role in cellular iron uptake [64].Toll-like Receptors (TLRs): Engagement with TLRs implies Gal-3’s potential in shaping immune responses [31,35].RAGE: Through interacting with Receptor for Advanced Glycation End Products (RAGE), Gal-3 can influence inflammation and oxidative stress responses [65].

Given this broad range of interactions, Gal-3 emerges as a versatile molecule impacting diverse biological processes from cell proliferation to immune responses. Its potential as a therapeutic target in diseases such as cancer, fibrosis, and inflammation underscores the importance of studying Gal-3 in the context of viral infections.

## 3. Galectin-3 in Specific Viral Diseases

The involvement of Gal-3 in various viral infections highlights its importance in host–pathogen interactions. Here, we outline the role of Gal-3 in specific viral diseases.

### 3.1. SARS-CoV-2 Virus Infection and Galectin-3

The COVID-19 pandemic, caused by the *SARS-CoV-2* virus, has affected millions of people worldwide [66]. The severity of the disease varies, with some individuals developing critical conditions such as acute respiratory distress syndrome (ARDS), cytokine storm, and thromboembolic complications [67]. Identifying reliable prognostic markers and therapeutic targets is crucial for improving patient outcomes [68]. One potential candidate is Gal-3, a protein involved in inflammation, lung fibrosis, and thrombogenicity [69,70].

#### 3.1.1. The Role of Galectin-3 in Modulating Immune Response and Inflammation in COVID-19

Galectin-3 is a multifunctional protein that plays a crucial role in various biological processes, including inflammation, cell adhesion, and immune modulation [10]. In the context of *SARS-CoV-2* infection, Gal-3 has been implicated in modulating the host immune response, potentially contributing to the severity of COVID-19. Gal-3’s pro-inflammatory effects include promoting inflammation through stimulating the production and release of pro-inflammatory cytokines, such as TNF-α, IL-1β, and IL-6 [13]. Elevated levels of these cytokines have been associated with severe COVID-19 cases, where patients may experience a “cytokine storm”, leading to ARDS and multi-organ failure [33,67].

Galectin-3 has also been shown to activate the NLRP3 inflammasome, a critical component of the innate immune response that detects and responds to pathogens [16]. Activation of the NLRP3 inflammasome by Gal-3 results in the production and release of pro-inflammatory cytokines, including IL-1β and IL-18, which contribute to the overall inflammatory response during *SARS-CoV-2* infection [22]. This inflammasome activation may potentiate the severity of COVID-19, particularly in patients with pre-existing conditions or comorbidities.

Recent studies have shown a significant association between elevated Gal-3 levels and COVID-19 severity [22]. Patients with critical COVID-19 have been found to have higher serum levels of Gal-3, along with increased proinflammatory cytokines (IL-1β, TNF-α, IL-12) and chemokine C-C chemokine receptor type 5 (CCR5) expression in T cells [22]. These findings suggest that Gal-3 may contribute to the acquired proinflammatory immune response and intensify the innate pro-inflammatory immune response in the lungs, making it a valuable marker for predicting disease severity.

#### 3.1.2. Potential Role of Human Galectin-3 in *SARS-CoV-2* Viral Adherence and Entry

The N-terminal domain of the SARS-CoV-2 spike protein plays a crucial role in viral adherence and entry into host cells [16]. Recent studies have revealed that the NTD of *SARS-CoV-2* shares a significant degree of similarity with human Gal-3, a lectin involved in numerous biological processes, including inflammation and cell adhesion [13]. The structural similarities between the NTD of *SARS-CoV-2* and human Gal-3 suggest that they may share common interaction partners or have similar functional roles, potentially facilitating *SARS-CoV-2*’s ability to hijack cellular processes and enhance its infectivity [13].

#### 3.1.3. Galectin-3 as a Potential Biomarker for COVID-19 Severity and Prognosis

Galectin-3 is a potential biomarker for predicting COVID-19 severity, prognosis, and treatment response [71]. Patients with severe COVID-19 exhibit higher levels of Gal-3, which are associated with increased inflammation, cytokine release, ARDS, and complications such as thrombosis, organ failure, and lung fibrosis [72]. Gal-3 levels correlate with inflammatory markers such as CRP, IL-6, and TNF-α, and monitoring these levels can help clinicians identify high-risk patients and guide clinical decision-making [73]. Galectin-3 also correlates with markers of thrombogenicity and clinical disease severity, suggesting its usefulness in assessing disease severity and hypercoagulability in COVID-19 patients [69]. In Table 2, we provide an overview of the diverse applications and implications of Gal-3 as a biomarker across different viral infections.

Galectin-3 is involved in various aspects of COVID-19 pathogenesis, including promotion of inflammation, dysregulation of immune response, fibrosis and tissue remodeling, and endothelial dysfunction and thrombosis [69,74]. Understanding the role of Gal-3 in COVID-19 pathogenesis can inform the development of targeted therapies and management strategies, potentially improving patient outcomes and reducing the global burden of this devastating pandemic.

#### 3.1.4. Comparative Roles and Interactions of Galectin-3 and Galectin-3-Binding Protein in *SARS-CoV-2* Infection

Galectin-3 and Galectin-3-binding protein (Gal-3BP) both play significant roles in viral infections [14]. Galectin-3-binding protein is a multifunctional glycoprotein encoded by the *LGALS3BP* gene, with a wide distribution across various cell types and biological fluids [75]. With its three functional domains, Gal-3BP engages in intricate cell-to-cell and cell-to-matrix interactions [14]. It is part of a broad spectrum of biological processes, including immune response, inflammation, and tumor progression [14]. Gal-3BP’s relationship with Gal-3 is particularly noteworthy, as it forms an important axis in various immune and inflammatory processes [76]. Elevated levels of Gal-3BP are found in a range of pathological conditions, such as cancer, viral infections, and autoimmune disorders, often correlating with adverse clinical outcomes [14].

In the context of viral infections, Gal-3 and Gal-3BP share some similarities but also exhibit important differences in their roles [14]. Both proteins have been associated with unfavorable outcomes in certain viral diseases, including *SARS-CoV-2* infection [77,78]. They are both involved in the innate immune response to viral infections, stimulating the production of interferon and pro-inflammatory cytokines. Their interaction is believed to modulate the expression of IL-6, a cytokine which plays a critical role in the immune response to viral infections [14].

However, while Gal-3 primarily facilitates viral entry and replication, Gal-3BP appears to exhibit antiviral activities, limiting virus expression and replication [14]. This difference suggests that while Gal-3 enhances virus propagation, Gal-3BP participates in the immune system’s defense against viral infections [14].

In the context of *SARS-CoV-2* infection, the levels of both Gal-3 and Gal-3BP are elevated, with higher levels associated with disease severity. They are both thought to contribute to the pathogenesis of COVID-19, particularly through their involvement in the excessive inflammatory response observed in severe cases. However, Gal-3BP has been specifically associated with worse clinical outcomes, whereas Gal-3 levels correlate with disease severity [14].

Interestingly, the interaction between Gal-3 and Gal-3BP has been linked to the development of pulmonary fibrosis in severe cases of COVID-19. This is because Gal-3, through its interaction with Gal-3BP, enhances Transforming Growth Factor beta (TGF-β) signaling, a key driver of fibrosis. This suggests that targeting the Gal-3/Gal-3BP axis could be a potential therapeutic approach for preventing or treating pulmonary fibrosis in severe COVID-19 cases [14].

Nevertheless, the precise roles of Gal-3 and Gal-3BP in viral infections, including *SARS-CoV-2*, are still not fully understood, and further research is needed to elucidate their respective contributions and the potential therapeutic implications.

### 3.2. Human Immunodeficiency Virus (HIV) Infection and Galectin-3

The *human immunodeficiency virus* is a retrovirus that attacks the immune system, leading to a progressive decline in its function. Over time, this decline can result in acquired immunodeficiency syndrome (AIDS), a severe condition characterized by a weakened immune system and increased vulnerability to opportunistic infections and malignancies [79].

Galectin-3 is a protein that has been implicated in various aspects of *HIV* infection, including viral entry, infection, dissemination, and pathogenesis.

#### 3.2.1. Galectin-3 in HIV-1 Entry and Infection

Research has demonstrated that Gal-3 can bind to *HIV-1* gp120 and CD4 proteins, enhancing viral entry and infection. The carbohydrate recognition domain of Gal-3 interacts with *HIV-1* gp120 or CD4 molecules, triggering the formation of Gal-3 oligomers, which facilitate infection [15]. Galectin-3 has been found to enhance *HIV-1* entry and infection up to 20-fold, suggesting a critical role in the early stages of viral infection [40]. Additionally, differences in Gal-3 expression among various *HIV-1* subtypes have been observed, implying that further investigation is necessary to understand the specific roles of galectin-3 in different viral strains [80].

#### 3.2.2. Galectin-3 Expression and *HIV-1* Tat Protein

The *HIV-1* Tat protein is a crucial factor in viral replication, playing a significant role in the transactivation of the *HIV-1* promoter and the efficient transcription of the viral genome [81]. Besides regulating viral gene expression, Tat also modulates host cell gene expression, promoting cell survival and proliferation and suppressing immune responses [82]. The interaction of Tat with host factors, such as specificity protein 1 (Sp1), is crucial in regulating both viral and host gene expression and underscores its importance in the pathogenesis of *HIV-1* infection [83]. Notably, the expression of Gal-3 is upregulated in *HIV-1*-infected cells, particularly during the early stages of infection [84]. Tat has been shown to induce Gal-3 expression through the transactivation of Sp1-rich regulatory sequences upstream of the Gal-3 gene [41]. The observed interaction between Sp1 and Tat in *HIV-1*-infected cells suggests a potential role for Tat protein in the regulation of Gal-3 expression. Therefore, the upregulation of Gal-3 expression in Tat-expressing cells highlights the potential of viral infection to induce the expression of this protein, which may contribute to the pathogenesis of HIV infection [41].

#### 3.2.3. Galectin-3 and *HIV-1* Dissemination

Galectin-3 plays a crucial role in the process of *HIV-1* viral budding, which is the final stage of the viral replication cycle [42]. This stage involves the release of newly formed virus particles from infected host cells, contributing significantly to the spread of *HIV-1* infection within the host and the progression of the disease [43]. Furthermore, Gal-3 is highly expressed in exosomes derived from *HIV-1*-infected dendritic cells, compared to those from uninfected cells [40]. This finding suggests that Gal-3 may facilitate *HIV-1* infection and dissemination. However, the role of Gal-3 in exosomes derived from *HIV-1* infected T cells appears to be cell-type specific, warranting further investigation [40].

Cell-to-cell transmission of *HIV-1* is an efficient mode of viral dissemination, allowing the virus to bypass extracellular defenses and evade host immune responses. Galectin-3 has been implicated in facilitating this process, contributing to the spread of the virus within the host [43]. Virological synapses are specialized cell–cell junctions that enable direct transfer of *HIV-1* particles from infected cells to uninfected target cells [85]. Galectin-3, with its ability to bind glycans on cell surfaces, may promote the formation of these synapses through stabilizing cell–cell contacts and enhancing adhesion between infected and uninfected cells [39]. This interaction could lead to more efficient viral transmission and faster dissemination within the host. Additionally, the presence of Gal-3 in exosomes released from *HIV-1* infected cells, particularly dendritic cells, may also contribute to cell-to-cell transmission [40]. The increased expression of Gal-3 in exosomes derived from infected cells may facilitate the transfer of viral particles and promote infection in recipient cells [40].

#### 3.2.4. Galectin-3 and *HIV-1*-Infected Macrophage Cell Death

Galectin-3 has been shown to induce cell death in *HIV-1*-infected macrophages, which could potentially contribute to the eradication of the virus [86]. The mechanism of Gal-3-induced cell death appears to involve glycosylation changes in the Gal-3 receptor and is caspase-independent, not relying on receptor-interacting protein kinase 1 (RIPK1)- or RIPK3-dependent necroptosis [86]. Furthermore, the study found that Endo G levels were significantly increased in the nucleus and decreased in the cytoplasm in Gal-3-treated cells, suggesting that Gal-3 may act as a novel apoptosis-inducing agent in response to *HIV-1* infection [86].

#### 3.2.5. Galectin-3 and HIV-Associated Pathogenesis

Elevated expression of Gal-3 has been implicated in *HIV*-associated pathogenesis, potentially contributing to the development of malignancies such as Kaposi’s sarcoma [87]. Gal-3 has been associated with tumor cell growth, and its upregulation may be linked to the proliferative response observed in *HIV*-infected individuals [12,87]. Therefore, understanding the intricate interplay between Gal-3 expression, *HIV* infection, and disease progression is critical for developing novel therapeutic strategies to improve the management of *HIV*-infected patients.

### 3.3. Influenza A Virus Infection and Galectin-3

*Influenza A virus* (*IAV*) is a member of the *Orthomyxoviridae* family and a major cause of respiratory illness worldwide. Seasonal outbreaks and occasional pandemics of *IAV* infection contribute to significant morbidity, mortality, and economic burden [88]. *Influenza A virus* is an enveloped, negative-sense, single-stranded RNA virus with a segmented genome [89]. The virus is classified into subtypes based on the combination of two surface glycoproteins, hemagglutinin (HA) and neuraminidase (NA) [90]. H5N1 is a highly pathogenic avian influenza virus that can cause severe disease in humans, characterized by high fever, pneumonia, and acute respiratory distress syndrome [91].

#### 3.3.1. Galectin-3 and *Influenza A Virus* Infection

Galectin-3 promotes *IAV* entry into host cells through stabilizing the interaction between viral HA and host sialic acid receptors on airway epithelial cells [18]. This process enhances the efficiency of *IAV* attachment and entry into host cells, promoting viral infection. During *IAV* infection, viral neuraminidase cleaves sialic acid residues from host cell surface glycoproteins, exposing the underlying β-galactoside-containing glycoconjugates [18]. Galectin-3 can then bind to these exposed glycoconjugates, further facilitating *IAV* attachment and entry into host cells [18].

#### 3.3.2. Promoting Viral RNA Synthesis

Galectin-3 plays a significant role in promoting viral RNA synthesis during *IAV* infection through enhancing the nuclear import of viral ribonucleoproteins (vRNPs), which are responsible for viral genome transcription and replication [19]. Furthermore, Gal-3 increases the activity of RNA-dependent RNA polymerase (RdRp), which is essential for IAV transcription and replication [19]. This increased RdRp activity contributes to the overall efficiency of viral RNA synthesis during *IAV* infection. Galectin-3 is also suggested to interact with heterogeneous nuclear ribonucleoproteins (hnRNPs), such as hnRNP A2/B1 and hnRNP L, which are known to upregulate *IAV* transcription and replication, further promoting viral RNA synthesis [19].

#### 3.3.3. Modulating Host Immune Response against *IAV*

Gal-3 plays a critical role in initiating and amplifying host inflammatory responses in the course of IAV infection [92]. In animal models, Gal-3 knockout mice exhibit reduced lung inflammation and lower levels of pro-inflammatory cytokines such as IL-1β following H5N1 influenza infection [44].

Galectin-3 interacts with the NLRP3 inflammasome, facilitating its assembly and oligomerization [36]. This interaction enhances the production of IL-1β from macrophages and contributes to the inflammatory response and lung damage observed in *IAV*-infected individuals [44].

Galectin-3 modulates the expression of antiviral genes, including those involved in the interferon (IFN) response [14]. Treatment with aloe-emodin, an anthraquinone derivative, has been reported to upregulate Gal-3 expression in infected cells, leading to increased expression of antiviral genes such as IFN-β, IFN-γ, protein kinase R (PKR), and 2′,5′-OAS via the JAK/STAT signaling pathway [93].

*Influenza A virus* infection can lead to secondary bacterial infections, such as *Streptococcus pneumoniae*, which can exacerbate disease severity [94]. Galectin-3 has been implicated in modulating the immune response during *IAV* and *S. pneumoniae* co-infection [19]. Studies have shown that Gal-3 expression is upregulated during co-infection and that it can bind to the epithelial cell surface, activating signaling pathways that regulate pro-inflammatory cytokine expression. Although this can result in hypercytokinemia, Gal-3 downregulation of SOCS1 expression enables a strong cytokine response, which is necessary for an effective antimicrobial defense against *S. pneumonia* [21].

### 3.4. Galectin-3 and Hepatitis Virus Infections

*Hepatitis B* (*HBV*) and *hepatitis C* (*HCV*) *viruses* are major global public health concerns, causing acute and chronic liver diseases that can lead to severe complications such as cirrhosis and hepatocellular carcinoma [95]. Galectin-3, a protein involved in inflammation and fibrosis, has been implicated in the progression of both *HBV* and *HCV* infections [23]. This section will explore the role of Gal-3 in these hepatitis virus infections and its potential as a biomarker for disease progression and treatment outcomes.

#### 3.4.1. Galectin-3 and *Hepatitis B Virus* Infections

Galectin-3 is known to stimulate cytokine and chemokine production through interacting with macrophages and may play a significant role in sustaining *HBV* replication and contributing to chronic *HBV* infection [23]. It can promote inflammation, leading to the progression of chronic *HBV* infection and liver fibrosis development. Studies have shown that serum Gal-3 levels are significantly higher in children with chronic hepatitis B (CHB) compared to inactive *HBV* carriers and healthy controls [23]. This finding suggests that Gal-3 may serve as a valuable biomarker for liver inflammation and the chronic phase of CHB in children, providing insight into the disease’s progression.

#### 3.4.2. Galectin-3 and *Hepatitis C Virus* Infections

Galectin-3 has also been implicated in *HCV* infection. Research comparing serum levels of Gal-3 in *HCV* patients before and after treatment with direct-acting antivirals (DAAs) has shown that *HCV* infection increases serum Gal-3 levels, which may contribute to inflammation and fibrosis progression [96]. However, successful DAA treatment reduces serum Gal-3 levels, suggesting a potential role for Gal-3 in monitoring *HCV* treatment outcomes [96].

Additionally, in patients with end-stage renal disease (ESRD) and *HCV* infection, Gal-3 has been shown to have hepatoprotective properties [48]. Elevated systemic levels of IL-6 and Gal-3 in ESRD *HCV*+ patients may counteract ongoing proinflammatory processes and downregulate chronic inflammation. Thus, Gal-3 might have a protective function in patients with HCV and ESRD, providing new insights into the complex relationship between HCV infection and ESRD [48].

### 3.5. Galectin-3 and Herpes Simplex Virus

Galectin-3 has been implicated in the attachment and entry of the *herpes simplex virus* (*HSV*) during ocular infections, as well as in herpetic allodynia, a type of pain induced by *HSV-1* infection [17,97]. This section will discuss the role of Gal-3 in HSV infections and its potential as a therapeutic target and biochemical marker.

#### 3.5.1. Galectin-3 in *HSV* Attachment and Entry

Galectin-3, a chimera lectin known to form pentamers upon binding to multivalent ligands, has been identified as a binding mediator during *HSV* infection, particularly in corneal keratinocytes [17]. It facilitates *HSV-1* binding and infection of these cells through directly interacting with the virus. The apical surface of the polarized epithelium contains a highly glycosylated glycocalyx, which serves as a barrier to virus infection [17]. Transmembrane mucins can bind to Gal-3 and restrict *HSV-1* infection in a concentration-dependent manner via providing steric hindrance [17]. Galectin-3’s interaction with transmembrane mucins is crucial in limiting *HSV-1* interactions with the glycocalyx, highlighting its potential as a therapeutic target for reducing herpesvirus infectivity.

#### 3.5.2. Galectin-3 and Herpetic Allodynia

Galectin-3 has been found to play a significant role in herpetic allodynia, a type of pain induced by *HSV-1* infection [97]. Its presence is markedly increased in the spinal cord, specifically in the superficial dorsal horn in cells positive for macrophage and microglia markers. A deficiency in Gal-3 significantly reduces herpetic allodynia, suggesting its contribution to the condition through infiltrating macrophages and/or resident microglia in the spinal dorsal horn, indicating that Gal-3 may be a new therapeutic target for the treatment of herpes zoster-associated pain [97].

Infiltrating monocytes and macrophages, positive for Iba-1 and F4/80 markers, are likely the source of Gal-3 in the spinal dorsal horn [97]. These cells may contribute to herpetic allodynia through Gal-3-dependent and -independent mechanisms, with Gal-3 reported to increase the expression and phosphorylation of extracellular signal-regulated kinase 1/2 (ERK1/2), a pathway known to participate in neuropathic pain states [97]. Serum Gal-3 levels have been identified as a potential biochemical marker for herpes zoster neuralgia (HZN) and postherpetic neuralgia (PHN) occurrence [98].

### 3.6. Galectin-3 and Cytomegalovirus

*Cytomegaloviruses*, such as *Human Cytomegalovirus* (*HCMV*) and *Murine Cytomegalovirus* (*MCMV*), are members of the *Betaherpesvirinae* subfamily and establish lifelong latent infections in their hosts [99]. *HCMV*, also known as human herpesvirus 5 (*HHV-5*), can cause severe complications in immunocompromised patients and is the most common cause of congenital viral infections [100]. *MCMV* serves as an essential model for understanding the biology, immune response, and pathogenesis of *HCMV* due to similarities in viral structure, replication, and host interactions [101].

Recent research has focused on the role of host molecules such as Gal-3 in the interaction between *MCMV* and its host cells [49]. A study investigating Gal-3’s role in MCMV-induced hepatitis found that Gal-3 plays a protective role in this condition [49]. Galectin-3-deficient mice experienced more severe liver damage, increased serum levels of alanine aminotransferase (ALT), and higher virus titers compared to normal mice. Treatment with exogenous Gal-3 alleviated *MCMV*-induced liver damage, further supporting its protective role [49].

The study indicates that elevated Gal-3 expression in *MCMV*-infected livers protects hepatocytes from TNF-α-facilitated apoptosis and necroptosis, consequently reducing liver damage in *MCMV*-induced hepatitis [49]. These findings suggest a protective role for Gal-3 in *MCMV*-induced liver damage, which could potentially be therapeutically relevant in future studies and treatments for *CMV*-induced hepatitis.

### 3.7. The Role of Galectin-3 in Other Viral Infections

The role of Gal-3, a beta-galactoside-binding lectin, in various viral infections has garnered significant attention due to its involvement in multiple aspects of virus–host interactions. Gal-3 has been implicated in the pathogenesis of *Kaposi’s sarcoma-associated herpesvirus* (*KSHV*), *adenovirus*, *enterovirus type 71* (*EV71*), *minute virus of mice* (*MVM*), *coxsackievirus B3* (*CVB3), human T-lymphotropic virus type 1* (*HTLV-1*), and *Junín virus* (*JUNV*). In this section, we discuss the recent findings on the role of Gal-3 in these viral infections as well as the potential therapeutic strategies that could be developed through targeting Gal-3 or modulating its function. Understanding the mechanisms and roles of Gal-3 in viral infections could pave the way for new treatment and prevention options for various viral diseases.

#### 3.7.1. *Kaposi’s Sarcoma-Associated Herpesvirus*

*Kaposi’s sarcoma-associated herpesvirus*, or *human herpesvirus 8* (*HHV-8*), is a gamma-herpesvirus linked to malignancies such as Kaposi’s sarcoma (KS), primary effusion lymphoma, and multicentric Castleman’s disease, especially in immunocompromised individuals [102]. Galectin-3 has been implicated in *KSHV* infection and Kaposi’s sarcoma pathogenesis [87]. Galectin-3 expression is downregulated in *KSHV*-infected dermal microvascular endothelial cells (DMVEC), suggesting its potential role as a biomarker for early detection and progression of *KSHV* infection. Research has shown that *KSHV* viral genes, such as *viral FLICE inhibitory protein* (*vFLIP*) and *Latency-Associated Nuclear Antigen* (*LANA*), suppress Gal-3 expression, possibly promoting laminin–cancer-cell interactions and leading to a more invasive tumor phenotype [87].

#### 3.7.2. *Adenovirus*

*Adenoviruses* are non-enveloped, double-stranded DNA viruses causing various infections, including respiratory, gastrointestinal, and ocular diseases [103]. Galectins, such as Gal-3, have been found to play a role in virus–host interactions during *adenovirus* infections [84]. Galectin-3 has shown both suppressive and beneficial effects in different scenarios, suggesting a complex role in host–virus interactions. In a model of *adenovirus*-induced lung injury, mice lacking Gal-3 experienced reduced lung fibrosis and β-catenin activation, implicating a connection to TGF-β signaling [104]. Additionally, *adenovirus*-infected BV2-microglial cells showed increased Gal-3 secretion and upregulated IFN-γ levels [56]. These findings highlight Gal-3’s importance in adenovirus infections and suggest that modulating Gal-3 function could offer a potential therapeutic approach for treating adenovirus-induced diseases.

#### 3.7.3. *Enterovirus Type 71*

*Enterovirus type 71*, a single-stranded RNA virus, is a major causative agent of hand-foot-and-mouth disease (HFMD), primarily affecting children under 5 years old. *EV71* infections can also lead to severe complications such as aseptic meningitis, acute flaccid paralysis, and brainstem encephalitis [105]. Research has shown that Gal-3 plays a role in *EV71* replication and pathogenesis, with Gal-3 knockout host cells exhibiting significantly reduced viral loads [20]. This association suggests a potential role for Gal-3 in the development of *EV71* infections and underscores the need for further research to understand the mechanisms behind Gal-3’s involvement and the potential for targeted therapies that modulate Gal-3 function to treat or prevent severe EV71-associated complications.

#### 3.7.4. *Minute Virus of Mice*

The *minute virus of mice* is a small, non-enveloped, single-stranded DNA virus from the *Parvoviridae* family, known for its oncotropic properties, infecting and killing cancer cells [106]. A study examining Gal-3 and N-acetylglucosaminyltransferase V (Mgat5) in *MVM* infection found that both are crucial for efficient cell entry and infection, with cancer cells expressing higher levels of Gal-3 being more susceptible to *MVM* infection [107]. These findings suggest a role for Gal-3 in *MVM* oncotropism, highlighting the need for further research to understand Gal-3’s involvement in *MVM* infection and potentially develop novel cancer therapies using oncolytic viruses such as *MVM*.

## 4. Therapeutic Potential of Targeting Galectin-3

Galectin-3 is a protein that plays a crucial role in various aspects of viral infections, making it an attractive target for antiviral therapy [14,108]. This section explores the therapeutic potential of targeting Gal-3 in viral infections.

### 4.1. Small Molecule Inhibitors of Galectin-3

Small molecule inhibitors targeting Gal-3 have shown potential as antiviral agents in preclinical studies, reducing viral replication and alleviating virus-induced inflammation [24]. The Gal-3 inhibitor TD139 has demonstrated significant reductions in lung fibrosis and ß-catenin activation in animal models of pulmonary fibrosis and is now in phase IIb trials for the treatment of idiopathic pulmonary fibrosis and COVID-19 [109]. A clinical trial of the inhaled Gal-3 inhibitor GB0139 for the treatment of COVID-19 found that it reduced Gal-3 concentrations and other key prognostic biomarkers associated with severe disease, showing potential to reduce the severity of systemic inflammation and cytokine excess in patients with COVID-19 [110].

### 4.2. Monoclonal Antibodies Targeting Galectin-3

The use of monoclonal antibodies targeting Gal-3 has shown promise as a potential therapeutic option [111]. These antibodies could have the ability to inhibit viral entry and regulate immune responses through blocking the interaction between Gal-3 and its binding partners, which include viral glycoproteins and host cell receptors [14]. This interference can effectively impede viral infection and mitigate the pathogenesis caused by the virus.

### 4.3. RNA Interference Techniques for Galectin-3 Silencing

RNA interference techniques, such as small interfering RNA (siRNA) and short hairpin RNA (shRNA), represent promising therapeutic options in the realm of medical science, as they can effectively silence Gal-3 expression and inhibit viral infections [112]. Through specifically targeting Gal-3 mRNA, these approaches could potentially reduce Gal-3 protein levels and disrupt its functions in viral infection. Since there have been no clinical trials conducted on these techniques, their potential for the development of novel therapies for viral infections, including those associated with Gal-3, cannot be overlooked.

## 5. Challenges and Future Directions

Despite the growing evidence supporting the pivotal role of Gal-3 in viral infection, several challenges and future directions must be addressed to further elucidate its multifaceted functions and therapeutic potential. In this section, we will discuss the current limitations in understanding Galectin-3’s involvement in host–pathogen interactions and explore possible avenues for future research. In Table 3, we present a comprehensive list of key topics and questions that warrant further investigation, which will ultimately contribute to a more in-depth understanding of Galectin-3’s roles and its potential as a therapeutic target in the context of viral infections.

### 5.1. Delineating the Molecular Mechanisms of Galectin-3 in Viral Infection

A comprehensive understanding of the molecular mechanisms through which Gal-3 mediates host–pathogen interactions is crucial for the development of targeted therapies. Future research should focus on identifying the specific domains and binding partners of Gal-3 that are involved in viral entry, replication, and immune modulation.

### 5.2. Identifying Galectin-3’s Role in Various Viral Infections

The role of Gal-3 in different viral infections remains incompletely understood. Future studies should investigate its involvement in a broader range of viruses and explore possible differences in its functions depending on the specific virus and host factors.

### 5.3. Evaluating the Therapeutic Potential of Galectin-3 Targeting

While several therapeutic strategies targeting Gal-3 have shown promise in preclinical studies, their safety and efficacy in clinical trials remain to be determined [113]. Further research should focus on optimizing these strategies and evaluating their potential for treating various viral infections in human patients.

### 5.4. Assessing the Potential Side Effects of Galectin-3 Inhibition

As Gal-3 plays essential roles in various physiological processes, inhibiting its function may lead to unintended consequences. Future studies should investigate the potential side effects of Gal-3 targeting and develop strategies to minimize any adverse outcomes.

### 5.5. Investigating the Role of Galectin-3 in the Context of Host Immunity

The complex interplay between Gal-3 and the host immune system remains to be fully elucidated. Future research should focus on understanding how Gal-3 modulates immune responses during viral infection and whether targeting Gal-3 can enhance antiviral immunity without exacerbating inflammation or causing immune-mediated damage.

### 5.6. Enhancing Understanding of Galectin-3 and Viral Interactions: A Critical Perspective

While existing research provides valuable insight into the diverse roles Gal-3 plays in various viral diseases, a more nuanced understanding of its specific interactions with viral entities is needed. Such understanding will likely be crucial in elucidating the intricate mechanisms underlying Gal-3’s involvement in the viral infection process. Hence, detailed investigations should be conducted to further uncover these specific interactions and their implications. Here, we suggest several key areas for future research to improve our understanding of these crucial interactions:

1. Molecular Mechanisms: Identifying the exact molecular mechanisms through which Gal-3 interacts with viruses can provide critical insights. This could involve studying its interaction with viral surface glycoproteins, and how it influences the viral entry, replication, or release.

2. Host–Virus Interactions: Galectin-3’s role in modulating the host immune response to viral infections needs further exploration. Studies could focus on how Gal-3 influences antiviral immunity, including its impact on innate immune responses and its potential role in triggering or suppressing inflammation during viral infections.

3. Therapeutic Implications: The potential of Gal-3 as a therapeutic target in viral infections is an area ripe for exploration. In vitro and in vivo studies that investigate the effects of Gal-3 inhibition or modulation on the course of viral infections could be valuable.

4. Virus-specific Studies: Investigations into how Gal-3 interacts with different types of viruses can provide disease-specific insights. Some viruses might exploit Gal-3 to enhance their infection, while in other cases, Gal-3 might play a protective role for the host.

5. Clinical Studies: Epidemiological studies could investigate the correlation between Gal-3 levels and the susceptibility, severity, or prognosis of viral infections in human populations.

Through addressing these areas, future research can provide a more complete picture of how Galectin-3 influences viral infections and inform the development of novel therapeutic strategies.

## 6. Conclusions

Galectin-3 plays a crucial role in various aspects of virus–host interactions, including viral entry, replication, and immune response modulation. Its involvement in a wide range of viral infections highlights its importance as a potential therapeutic target and biomarker for disease progression and treatment outcomes. Further research into the mechanisms underlying Gal-3’s involvement in these viral infections is necessary to develop targeted therapies that modulate Gal-3 function to treat or prevent severe complications associated with these diseases. Understanding the complex roles of Gal-3 in viral infections may lead to the development of novel strategies for managing and preventing various viral diseases, ultimately improving patient outcomes and reducing the global burden of viral illnesses.

## Figures and Tables

**Figure 1 ijms-24-09617-f001:**
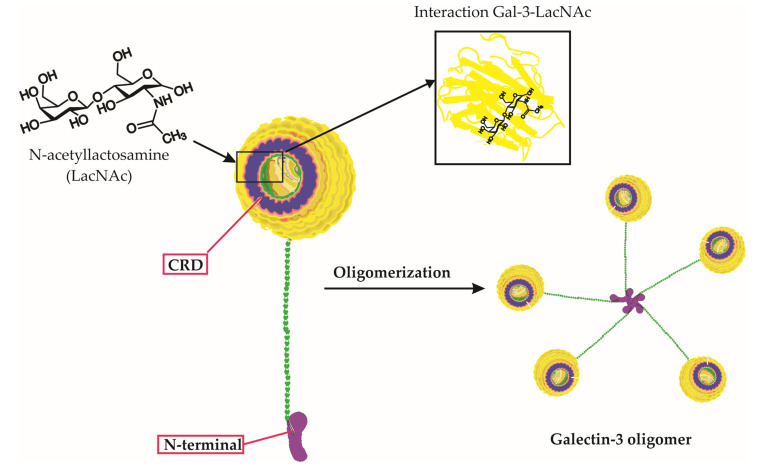
Galectin-3 Structure, Carbohydrate Binding, and Oligomerization. This figure presents an in-depth look at the Galectin-3 (Gal-3) molecule, emphasizing its unique chimeric structure, the specificity of its carbohydrate interactions, and the process of oligomerization. The three-dimensional representation of Gal-3 displays its distinctive carbohydrate recognition domain (CRD) and N-terminal tail. Notably, the figure highlights the binding of specific carbohydrate molecules, including lactose and N-acetyllactosamine (LacNAc), to the CRD of Gal-3. Furthermore, it depicts the oligomerization of Gal-3 molecules, wherein the N-terminal tails interact with each other to form complex higher-order structures. This interaction of Gal-3 and LacNAc within the CRD, along with its overall structure and carbohydrate binding capability, underscores Galectin-3’s critical roles in various biological processes.

**Figure 2 ijms-24-09617-f002:**
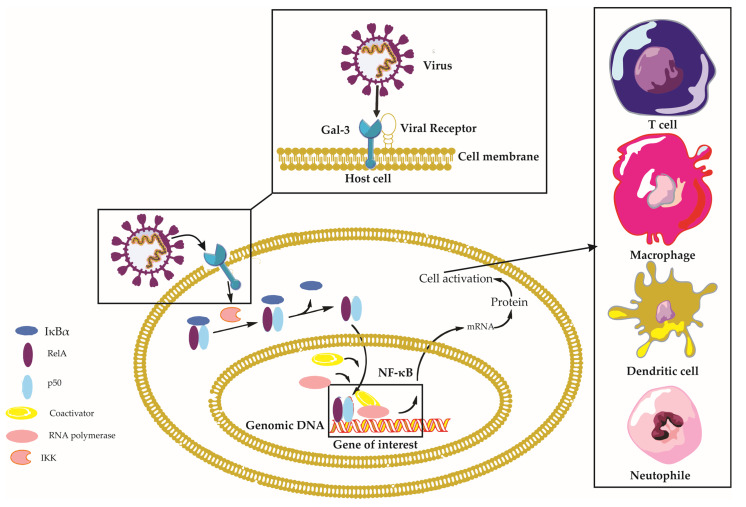
Galectin-3 in Viral Infections: A Comprehensive Overview. This figure depicts the intricate interplay between Galectin-3 (Gal-3), viral particles, and host cells during viral infections, emphasizing the activation of the Nuclear Factor Kappa B (NF-κB) signaling pathway. Viral particles interacting with Gal-3 on the host cell surface trigger various intracellular signaling pathways, including the NF-κB pathway. Normally, the inactive NF-κB complex—comprising RelA (p65), p50, and IκBα—is confined to the cytoplasm. The viral–Gal-3 interaction activates IκB kinase (IKK), which then phosphorylates IκBα. This phosphorylation signals IκBα for ubiquitination and subsequent proteasomal degradation, revealing the nuclear localization signal of the NF-κB dimer (RelA/p50), which translocates to the nucleus. Here, the NF-κB dimer binds to κB sites on DNA, thereby regulating the transcription of target genes with assistance from coactivators. This activity leads to diverse cellular responses, including immune response and inflammation. Concurrently, the figure illustrates immune cells, such as macrophages and T-cells, migrating towards the host cell in response to the Gal-3–virus interaction, underlining their essential roles in the immune response. Key components involved in this context include Nuclear Factor-kappa B (NF-κB), Receptor Enhancer of Nuclear Factor Kappa-B Ligand A (RelA), Inhibitor of Nuclear Factor Kappa-B Alpha (IκBα), NF-kappa-B p50 subunit (p50), I kappa B kinase (IKK), and Galectin-3 (Gal-3).

**Table 1 ijms-24-09617-t001:** Roles of Galectin-3 in viral infections.

Role	Description	Examples of Viruses Involved
Viral entry	Facilitates virus attachment and entry into host cells through binding to viral and host cell surface glycoproteins	*HIV* [15], *SARS-CoV-2* [16], *HSV* [17], *Influenza* [18]
Viral replication	Modulates key cellular signaling pathways that influence viral replication processes	*Influenza* [19], *Enterovirus* [20]
Immune response modulation	Regulates both innate and adaptive immune responses, influencing the balance between pro-inflammatory and anti-inflammatory responses	*Influenza* [21], *SARS-CoV-2* [22], *HBV* [23]
Potential therapeutic target	Development of novel Gal-3 inhibitors and their application in antiviral therapy	*SARS-CoV-2* [24]

**Table 2 ijms-24-09617-t002:** Galectin-3 as a biomarker in viral infections.

Viral Infection	Sample Type	Galectin-3 Expression Pattern	Clinical Implications
Influenza	Serum, bronchoalveolar lavage fluid	Elevated levels during infection	Prognosis, disease severity
*Human immunodeficiency virus*	Plasma, peripheral blood mononuclear cells (PBMCs)	Elevated levels in patients	Disease progression, immune dysfunction
*Hepatitis C virus*	Serum, liver tissue	Elevated levels in patients	Liver fibrosis, inflammation
*Severe acute respiratory syndrome coronavirus 2*	Serum, plasma, bronchoalveolar lavage fluid, lung tissue	Elevated levels in patients	Disease severity, prognosis, potential therapeutic target

**Table 3 ijms-24-09617-t003:** Areas for future research on Gal-3 in viral infections.

Research Area	Description
Mechanisms of Gal-3-mediated viral entry	Elucidate the molecular mechanisms through which Gal-3 facilitates virus attachment and entry into host cells
Role in viral latency	Investigate the potential involvement of Gal-3 in maintaining viral latency and reactivation processes
Immune response regulation	Unravel the complex regulatory mechanisms through which Gal-3 modulates host immune responses to viral infections
Development of novel Gal-3 inhibitors	Design, synthesis, and evaluation of new Gal-3 inhibitors with improved potency and selectivity for antiviral therapy
Clinical trials	Conduct clinical trials to assess the safety, efficacy, and potential side effects of Gal-3 inhibitors in antiviral therapy

## Data Availability

This study is a review article and does not generate any new datasets or involve the use of existing datasets. The authors have drawn upon and analyzed previously published research in the field, and all relevant sources have been cited within the manuscript. Readers interested in accessing the original data sources can refer to the cited references for further information.

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
