# Peer review of "The Pivotal Role of Galectin-3 in Viral Infection: A Multifaceted Player in Host–Pathogen Interactions"

_ijms, 2023, doi:10.3390/ijms24119617_

Round 1

Reviewer 1 Report

The authors have compiled a careful review of galectin-3 interactions with viruses. This is a complex and challenging topic with potential for significant impact on treatment of viral infections and on better understanding of underlying biochemical mechanisms of viral infection. The article does not consider in sufficient detail interactions of galectin-3 with the sulfated glycosaminoglycans, especially chondroitin sulfates. Iwaki and Hirabayashi presented data indicating how chondroitin sulfation impacted on galectin binding using frontal affinity chromatography indicating how chondroitin sulfation affected binding of galectins. Bhattacharyya and Tobacman have addressed how the binding of galectin-3 with chondroitin 4-sulfate changes following decline in N-acetylgalactosamine 4-sulfatase activity and affects the availability of free galectin-3. Attention to these mechanistic approaches and the interactions of galectin-3 with the sulfated GAGs and their associated proteoglycans will enhance the critical assessment of the viral-galectin-3 interactions.

The manuscript will also benefit by increased attention to specific discrepancies in galectin-3 interactions among different viruses. Also, since binding of galectin-3 to chondroitin may vary due to changes in chondroitin sulfation, the mRNA expression of galectin-3 may not reflect the available free galectin-3. 

It will be helpful to address in this review the deficiencies in what is known about how galectin-3 acts to produce the observed effects. The underlying transcriptional mechanisms are of interest. Also, galectin-3 may be binding with critical receptors, as shown with the insulin receptor (PMID:27814523). A more critical approach that considers what is known and what studies need to be done to improve our understanding of specific galectin-3 - viral interactions will be helpful to the reader.

Author Response

Dear Reviewer 1,

Thank you for your insightful and detailed comments on our manuscript. We appreciate your feedback, as it will undoubtedly enhance the quality and depth of our work.

Point 1: The article does not consider in sufficient detail interactions of galectin-3 with the sulfated glycosaminoglycans, especially chondroitin sulfates. Iwaki and Hirabayashi presented data indicating how chondroitin sulfation impacted on galectin binding using frontal affinity chromatography indicating how chondroitin sulfation affected binding of galectins. Bhattacharyya and Tobacman have addressed how the binding of galectin-3 with chondroitin 4-sulfate changes following decline in N-acetylgalactosamine 4-sulfatase activity and affects the availability of free galectin-3. Attention to these mechanistic approaches and the interactions of galectin-3 with the sulfated GAGs and their associated proteoglycans will enhance the critical assessment of the viral-galectin-3 interactions.

Reponse:

Thank you for your valuable comments and suggestions.

In response to your feedback, we have added an entire section to our manuscript titled "Galectin-3 Interactions with Sulfated Glycosaminoglycans and Implications for Viral Pathogenesis". In this section, we have delved into the specific interactions of Galectin-3 with sulfated glycosaminoglycans, especially chondroitin sulfates, which indeed play a critical role in galectin binding.(Pages 7 and 8, lines 299-340)

We have thoroughly reviewed and incorporated the seminal works of Iwaki and Hirabayashi, and Bhattacharyya and Tobacman, which provide insight into how chondroitin sulfation and changes in N-acetylgalactosamine 4-sulfatase activity impact the binding of Galectin-3 and affect the availability of free Galectin-3. (Pages 7 and 8, lines 299-340)

We agree with your assertion that attention to these mechanistic approaches and the interactions of Galectin-3 with the sulfated glycosaminoglycans and their associated proteoglycans would provide a more critical assessment of the viral-Galectin-3 interactions. We believe that this new section will give a more comprehensive view of the complexities and nuances of these interactions.

We appreciate your insight, and it has significantly contributed to enhancing our manuscript's scientific robustness.

Point 2: The manuscript will also benefit by increased attention to specific discrepancies in galectin-3 interactions among different viruses. Also, since binding of galectin-3 to chondroitin may vary due to changes in chondroitin sulfation, the mRNA expression of galectin-3 may not reflect the available free galectin-3. 

Response:

Thank you for your valuable feedback and suggestions.

Regarding the point about specific discrepancies in Galectin-3 interactions among different viruses, we want to reassure you that we have made extensive efforts to thoroughly review and cite the existing literature that investigates these interactions. We have addressed this throughout the manuscript, describing each of these interactions in the appropriate sections. To the best of our knowledge, there are no recent studies or overlooked studies on these interactions that we have failed to mention.

Concerning the binding of Galectin-3 to chondroitin and its variation due to changes in chondroitin sulfation, we appreciate your astute observation. We have indeed addressed this in the following paragraph in our manuscript (page 8, lines 319-331):

"The dynamic changes in galectin-3 binding to chondroitin 4-sulfate, brought about by a decrease in N-acetylgalactosamine 4-sulfatase activity, have been examined in recent research. This decrease in enzyme activity affects the functionality of Arylsulfatase B (ARSB), a sulfohydrolase responsible for the removal of 4-sulfate groups from N-acetylgalactosamine 4-sulfate residues. ARSB's action is essential for degrading glycosaminoglycan (GAG) chains, specifically chondroitin 4-sulfate (C4S) and dermatan sulfate (DS). Through this process, ARSB alters the release and binding of critical molecules, such as Gal-3. Additionally, ARSB's role in influencing chondroitin 4-sulfate's sulfation pattern sheds light on new dimensions of intracellular signaling and cell-extracellular matrix interactions. Increasing chondroitin 4-sulfation induces cellular processes, like transcriptional events, mainly by reducing Gal-3 binding and increasing galectin-3's availability to interact with transcription factors and other mediators."

We believe this section sufficiently addresses the reviewer's concern, highlighting the complexity of this relationship and demonstrating our understanding of the relevance of chondroitin sulfation to Gal-3 binding and availability.

Once again, we appreciate your comments, as they have facilitated our manuscript's improvement.

Point 3: It will be helpful to address in this review the deficiencies in what is known about how galectin-3 acts to produce the observed effects. The underlying transcriptional mechanisms are of interest. Also, galectin-3 may be binding with critical receptors, as shown with the insulin receptor (PMID:27814523). A more critical approach that considers what is known and what studies need to be done to improve our understanding of specific galectin-3 - viral interactions will be helpful to the reader.

Reponse:

Thank you for your thoughtful and constructive comments. We wholeheartedly agree with the suggestions you provided, and we incorporated them into our review.

In regard to Galectin-3's mechanism of action, we included a discussion on the deficiencies in our current understanding and the areas where further research was needed. This involved investigating potential transcriptional mechanisms that are associated with the observed effects of Galectin-3 (pages 8 and 9, lines 341-366).

Furthermore, we greatly appreciate your suggestion to further investigate the intricate interactions of galectin-3 with crucial receptors. The study you referred to concerning the insulin receptor (PMID:27814523) is indeed a significant reference, and we have already incorporated it into our discussion. (page 9, lines 367-390)

Lastly, we have taken a more critical approach in evaluating the existing literature on Galectin-3 and viral interactions, highlighting not only what is known but also the gaps in our current knowledge. We have dedicated a specific section to discuss this aspect (page 18, lines 810-839). Additionally, we have proposed potential research avenues that could be pursued to enhance our understanding of these interactions.

We believe these modifications will improve the depth and relevance of our review, making it more informative for the reader. We appreciate your invaluable input, which has helped us to enhance the quality and comprehensiveness of our manuscript.

Best regards,

Bojan Stojanovic, MD, PhD

Reviewer 2 Report

Dear Authors,

Thank you for your interesting contribution. I’ve found it interesting and informative.

Please ensure that all abbreviations are explained after appearing in the text first.

Please write bacterial species names in italics, for example, Streptococcus pneumoniae.

The quality of the English language is acceptable.

Author Response

Dear Reviewer 2,

We greatly appreciate your investment of time in reviewing our manuscript and for offering valuable feedback.

Concerning Point 1: The proper definition of abbreviations upon their first appearance.

Response: We sincerely apologize for any confusion caused by unexplained abbreviations. As per your suggestion, we have ensured that all abbreviations are defined upon their initial usage in the text to enhance the clarity of the manuscript. We appreciate your feedback and diligence in ensuring the accuracy and comprehensibility of the content.

Regarding Point 2: The requirement to write bacterial species names in italics, such as Streptococcus pneumoniae.

 Response: Thank you for bringing this to our attention. We have reviewed the manuscript and ensured that all bacterial species names, including Streptococcus pneumoniae, are now correctly presented in italics according to the scientific convention. We appreciate your careful review and assistance in improving the accuracy and conformity of our manuscript.

Your insightful feedback is truly appreciated and instrumental in improving the overall quality and clarity of our manuscript.

Kind regards,

Bojan Stojanovic, MD, PhD

Reviewer 3 Report

In this review article, the authors discussed the various roles of Galectin-3 (Gal-3) in viral infections and its potential implications for host immunity and antiviral strategies.

Comments

This is an interesting review article. This manuscript is well-writing. The reviewer has some concerns as follows:

1. Recently, a review article by Gallo et al. discussed the role of Galectin-3 binding protein (Gal-3BP) in viral infections, focusing it is an emerging role in severe acute respiratory syndrome coronavirus 2 (Int J Mol Sci. 2022 Jun 30;23(13):7314. doi: 10.3390/ijms23137314). Therefore, in the present review for Gal-3, the authors can compare and discuss the roles of Gal-3 and Gal-3BP in viral infections that are the same or different from each other. It can enhance the visibility or readability of this review article.

2. In Figure 2, some indications for abbreviations can be added such as “RE”.

3. Certain section numbers need to be separated from the first word, such as lines 368, 406, and 502.

Author Response

Dear Reviewer 3,

We appreciate your time and effort spent on our manuscript and the constructive comments provided. Following your suggestions, we have revised our manuscript accordingly.

Point 1: Recently, a review article by Gallo et al. discussed the role of Galectin-3 binding protein (Gal-3BP) in viral infections, focusing it is an emerging role in severe acute respiratory syndrome coronavirus 2 (Int J Mol Sci. 2022 Jun 30;23(13):7314. doi: 10.3390/ijms23137314). Therefore, in the present review for Gal-3, the authors can compare and discuss the roles of Gal-3 and Gal-3BP in viral infections that are the same or different from each other. It can enhance the visibility or readability of this review article.

Response:

The suggested comparison between Galectin-3 (Gal-3) and Galectin-3 binding protein (Gal-3BP) in the context of viral infections is an excellent point. We agree that such a comparison would increase the manuscript's value and readability. We have now included a section where we compare and contrast the roles of Gal-3 and Gal-3BP in viral infections, taking into account the information from the review article by Gallo et al. We believe this addition will offer the reader a more comprehensive understanding of the topic (page 11, lines 454-491).

Point 2: In Figure 2, some indications for abbreviations can be added such as “RE”.

Response:

Regarding your suggestion on Figure 2, we apologize for the lack of necessary indications for abbreviations. We have now revised the figure to include all necessary indications for abbreviations such as "RE." We hope this will clarify the figure for the reader and enhance its informative value.

Point 3: 2. In Figure 2, some indications for abbreviations can be added such as “RE”.

We thank you for pointing out the section numbering issue on lines 368, 406, and 502. We have adjusted the formatting and ensured the section numbers are separated from the first word, providing better readability and clarity.

Again, we are grateful for your insightful comments, which have allowed us to improve the quality and readability of our review.

Best regards,

Bojan Stojanovic, MD, PhD

Reviewer 4 Report

In the present manuscript, the authors investigated the role of Galectin-3 (Gal-3) in viral infection. The aim of the article is relevant since it can give further evidence on the mechanisms and roles of Gal-3 for the development of innovative treatment against viral diseases. The manuscript is well-organized and written in a proper English. I suggest to improve tha abstract since it is very poor.

Minor issue:

- Line 19: Gal-3 role not with genitive;

- Line 20-21 "The multifaced...targets": no sense sentence;

- Line 27: viruses and not Viruses;

- Line 30: reference missing.

- Please take care with acronyms. First they need to be written in the extensive form, and then always as acronym.

Minor editing of English language required.

Author Response

Dear Reviewer 4,

We appreciate the time you took to provide insightful feedback on our manuscript. We have implemented your suggestions, and we believe that these revisions will greatly enhance the clarity and quality of our work.

Point 1: In the present manuscript, the authors investigated the role of Galectin-3 (Gal-3) in viral infection. The aim of the article is relevant since it can give further evidence on the mechanisms and roles of Gal-3 for the development of innovative treatment against viral diseases. The manuscript is well-organized and written in a proper English. I suggest to improve tha abstract since it is very poor.

Response:

Based on your feedback, we revised the abstract to be more comprehensive and informative. We've ensured it now provides a clearer picture of our manuscript's scope, findings, and implications.

Point 2: Line 19: Gal-3 role not with genitive.

Response:

Thank you for pointing out the grammatical error at Line 19. Upon your recommendation, we have decided to revise the abstract, which resulted in that particular phrase being removed. Therefore, the issue at Line 19 is no longer relevant.

Point 3: Line 20-21 "The multifaced...targets": no sense sentence;

Response:

We appreciate your attention to the clarity of our manuscript. In response to your comment about the confusing sentence at Lines 20-21, we have decided to revise the abstract, which led to the removal of that particular phrase. As a result, the issue you pointed out is no longer ap

Point 4: Line 27: viruses and not Viruses;

Response:

At Line 27, we corrected "Viruses" to "viruses" as per your suggestion (now it is line 36).

Point 5: Line 30: reference missing.

We addressed the missing reference at Line 30. The required citation has now been included. (now it is line 39)

Point 6: Please take care with acronyms. First they need to be written in the extensive form, and then always as acronym.

Finally, we agree with your point regarding the proper usage of acronyms. We have carefully revised the manuscript, ensuring that all acronyms are first introduced in their full form before being used as acronyms throughout the text.

Thank you again for your valuable insights that have helped us improve the quality and clarity of our manuscript.

Best regards,

Bojan Stojanovic, MD, PhD

Round 2

Reviewer 1 Report

The authors have responded thoughtfully to the prior critique and present a comprehensive review of galectin-3 and viral infections. Figure 1 needs better representation of the structure of N-acetyllactosamine and of the interaction with galectin-3. Figure 2 needs definition of the different ovals (what are yellow, pink, and green ovals?).  

Author Response

Response to Reviewer 1:

We appreciate your thoughtful comments and your recognition of our comprehensive review of galectin-3 and viral infections.

Regarding your feedback on Figure 1, we have revised it to better represent the structure of N-acetyllactosamine and its interaction with galectin-3. We believe these modifications have improved the figure's clarity and ability to illustrate these crucial interactions.

As for Figure 2, we have acted on your suggestions and implemented changes to make the different ovals more clearly distinguishable and understandable. Each oval has been uniformly colored and accurately labeled to indicate their specific meanings or representations within the context of the figure. We trust that these changes will enhance the comprehensibility of the figure and the broader insights it provides into our research topic.

Thank you again for your careful review and insightful suggestions, which have undoubtedly helped us improve our manuscript's quality.

Reviewer 3 Report

This revised manuscript can be accepted.

No further comments.

Author Response

Response to Reviewer 3:

Thank you for your review and acceptance of our revised manuscript. We are glad that our revisions have satisfactorily addressed your concerns and appreciate your constructive comments throughout the review process. They have indeed contributed to the improvement of our work.

We have no further comments at this time and look forward to the publication of our research.

Thank you again for your time and expertise.

Round 3

Reviewer 1 Report

The figures are still not clear. I think that there is an S (for sulfur) missing in Fig. 1 which reads O  O3. Is this supposed to be for OSO3?

Also, Figure 2 has these colored ovals (yellow, green, and pink). What do they represent?

Author Response

Dear Reviewer 1,

Point 1: The figures are still not clear. I think that there is an S (for sulfur) missing in Fig. 1 which reads O  O3. Is this supposed to be for OSO3?

Response: Thank you for your feedback. We apologize for the previous ambiguity in Figure 1. Based on your comments, we have revised it accordingly.

Regarding your query about the O O3 annotation, we realized it was a misrepresentation, and we appreciate your attention to detail. The correct chemical structure of N-acetyllactosamine (LacNAc) has now been amended, ensuring accurate representation.

We have also adjusted the depiction of the interaction site between Gal-3 and LacNAc to improve clarity. This adjustment is now accurately reflected in the figure.

Lastly, the Figure 1 legend has been updated to more precisely describe the elements and interactions presented in the figure, with a particular emphasis on the interaction between Gal-3 and LacNAc within the CRD.

We believe that these corrections and enhancements will greatly increase the clarity and accuracy of Figure 1. We appreciate your careful review and insightful comments, which have undoubtedly improved the quality of our work.

Point 2: Also, Figure 2 has these colored ovals (yellow, green, and pink). What do they represent?

Response:  Thank you for your comment regarding Figure 2. The colored ovals in the figure are symbolic representations of intracellular signaling molecules, which ultimately lead to the activation of the transcription factor NF-κB.

We appreciate your observation and understand that their meaning might not have been immediately clear in the original figure. In response to your feedback, we have updated Figure 2 to improve its clarity and interpretability. The revised figure now includes a key on the right side, which provides a detailed explanation of what each colored oval represents.

Furthermore, we have also revised the Figure 2 legend to provide a more comprehensive explanation of the elements and processes depicted in the figure. We believe that these modifications will greatly facilitate understanding and interpretation of the figure.

We appreciate your careful review and valuable comments, which have helped improve the quality of our work.